# Certeze Village: The Dilemma of Traditional vs. Post-Modern Architecture in Țara Oașului, Romania

Iuliana Vijulie [1], Ana-Irina Lequeux-Dincă [1,*], Mihaela Preda [1], Alina Mareci [1], Elena Matei [1], Roxana Cuculici [1] and Ana-Maria Taloș [1,2]

[1] Faculty of Geography, University of Bucharest, 1.Blv. Nicolae Balcescu, 010041 Bucharest, Romania; iuliana.vijulie@g.unibuc.ro (I.V.); mihaela.preda@geo.unibuc.ro (M.P.); alina.mareci@unibuc.ro (A.M.); elena.matei@g.unibuc.ro (E.M.); roxana.cuculici@geo.unibuc.ro (R.C.); ana.talos@geo.unibuc.ro (A.-M.T.)

[2] Romanian Young Academy, University of Bucharest, 050663 Bucharest, Romania

\* Correspondence: ana.dinca@geo.unibuc.ro; Tel.: +0040-726-691-727

**Abstract:** The traditional Romanian village has recently seen unmistakable transformations. The import of architectural styles from EU countries and the need to modernise dwellings, combined with considerable legislative voids regarding the protection of the built-up heritage, have strongly modified traditional architecture and resulted in irremediable losses in terms of rural authenticity and landscape aesthetics. This study aims to analyse the need for preserving existing traditional architecture in Certeze village, which has been severely jeopardised by the import of post-modern elements. The perception of both locals and tourists on these aspects was evaluated using the survey method. Results outlined more conservative views from the older inhabitants who are still attached to traditional constructing styles, while younger respondents preferred the more modern houses. Most tourists also showed an increased interest in the traditional architecture and criticised the newer constructed buildings. The contrast between old and new, which at this point is ubiquitous in the area, remains an element of intergenerational negotiations and risks the diminishing of the cultural authenticity of Certeze even further.

**Keywords:** local community; traditional architecture; post-modernity; survey; Certeze; Romania





## 1. Introduction

Traditional architecture is often underestimated and becomes increasingly vulnerable in the current context, even though it is one of the essential advantages rural communities have, especially concerning tourism capitalisation. Numerous studies have emphasised the importance of its preservation and worldwide promotion [1,2]. The success of rural tourism sustainable policies in Romania hinges on national and regional efforts to maintain authentic architecture, traditions, and customs that define the uniqueness of its rural destinations [3–8].

The Țara Oașului ethnographical region is an example of a Romanian geographical area defined and promoted for its cultural authenticity, especially in terms of customs and architecture. Unfortunately, globalisation and modernity have somewhat obscured traditional architectural elements. The region chosen as a study area is an obvious example of this, as it unmistakably displays society's pressure in preferring comfort over traditional building methods and materials. Furthermore, day-to-day living needs are known causes for altering traditionally constructed buildings [9]. Traditional rural houses often carry increased maintenance costs and efforts due to their archaic materials and building tech-niques, which are more challenging to find and implement nowadays. In addition, the younger generation perceives older houses as being less functional and/or aesthetically less attractive [1]. Thus, rural inhabitants face a dilemma. On the one hand, they can now use new materials and styles when constructing their houses to increase their comfort according to current modern living standards. However, on the other hand, the desire

to keep traditions alive and/or ensure the consumption of authentic cultural heritage by tourists constrains them to maintain unaltered construction styles and materials in terms of the houses they live in.

The present paper includes an applied research study on the village of Certeze from Țara Oașului regarding an issue of great interest for the scientific community, local authorities, and tourism policy planners. It opens with a chapter dedicated to a literature review focusing on the topic in general and on a particular view of its relation to the study area, followed by a more detailed, illustrative analysis of the village of Certeze. The next chapter describes the field research methodology and the survey conducted for this study, which employed an approach less frequent in the specialised literature: obtaining an antithetical and complementary perspective of the endogenous (locals) and exogenous (tourists) population. This initiative, which is original and essential in the given context, can be helpful both nationally and internationally for comparative studies and can substantiate sustainable, integrated, and participative housing policies for other rural areas that enjoy both traditional architecture and significant tourism flows. Then, the article presents the main results and discussion, including recommendations, followed by the research conclusions.

### 1.1. Challenges for Traditional Architecture in Rural Areas

Globalisation is presently viewed as a defining phenomenon for society and has a lasting impact on outlining and modifying elements of local identity [10]. However, it often negatively impacts authenticity as it invites post-modernity to the detriment of tradition and brings somewhat of a decline in cultural diversity [11,12].

Traditional architecture is affected by social pressure as owners of traditional houses become more interested in their functionality, utility, comfort, and maintenance [13], and less in their aesthetics. As a result, the tangible cultural heritage, which is a manifestation of an old lifestyle, no longer corresponds to the present daily needs. This way, conserving inhabited traditional houses is currently a serious challenge [14]. Most times, high maintenance costs and strict regulations regarding heritage buildings entail numerous disadvantages for people living in traditional houses. As such, some (especially younger people) are tempted to abandon them in favour of more modern living conditions [15].

Tourism demands for the maintenance of local tangible cultural heritage, authentic rural landscapes, and traditional rural architecture. It is also an engine for economic diversification, development, and rural regeneration [16]. Agritourism, in particular, thrives from preserved rustic houses and authentic farms [17]. Tourists visiting rural areas seek something different from the urban areas they come from and traditional architecture, for example, remains an element of interest for them [18]. In fact, rural tourism becomes an opponent to modernising traditional houses, creates restrictions for planners in this field, and often leads to conflict situations in world heritage sites, but these challenges are not necessarily restricted to them [15,19–23]. When modern standards are to be included in the sustainable planning of rural tourism architecture, rural construction projects should still consider the region's natural and cultural landscape, and should blend into it through their materials and design [24,25]. This would avoid sterile architecture that is inconsistent with its area, which unfortunately occurs and results from local agritourism entrepreneurs' ignorance. Additionally, some modern accommodations for tourists in rural areas result in inflicting even more damage: although they respect the level of standardised services better than the old traditional houses, which would require profound transformations to reach the same level, they incorporate cheap construction materials and use a plain or, even worse, futuristic design inconsistent with their location [18].

At an international level, traditional architecture is an integral part of national heritage and preserving it is a priority because it describes the identity features of its people. It is also used as a primary attraction vector for tourism flows and, as such, is an economic development tool for the area. Therefore, countries such as the Netherlands, Spain, the Republic of North Macedonia, France, Slovakia, the United Kingdom, Germany, Croatia, and

Italy have made substantial efforts in preserving areas with traditional architecture [26–29]. The measures implemented in these countries aim to elaborate methodologies for restoration according to both usage and local customs to recapitalise/promote properties that incorporate traditional elements. Moreover, their good practice guidelines do the following: cover technical solutions for reducing the impact of implementing urban networks; establish types of construction materials permitted for building rehabilitation for each area of the country; promote training courses for workers involved in rehabilitating the built-up heritage; and organise awareness campaigns on the importance of ethnocultural heritage targeting the local population in their role as leading actor to ensure its survival and evolution [30–32]. In certain cases, depending on the historical period and the political regime [27], and mainly on the governmental involvement for heritage protection and rehabilitation [28,31,32], good practice guidelines were prescribed in administrative laws destined for certain categories of buildings and specific territorial units, although they often remained as architectural or engineering documents lacking regulatory frameworks.

Maintaining the architectural homogeneity of Romania's ethnocultural regions is a relatively recent endeavour. The initiative of establishing guidelines with recommendations for construction materials, styles, colours, and types of decorations only appeared due to pressures from civil society, local tourism entrepreneurs, and NGOs. Subsequently, the Order of Romanian Architects drew up guidelines for 40 distinct regions of the country. They included house layouts and architectural features that adapted to the local specificity and did not cause any visual discrepancies, such as in the case of the one dedicated to Ţara Oaşului [33]. This guiding document contains recommendations and illustrative examples of construction materials, dimensions, styles, and colors for houses, as well as for their main distinctive elements (e.g., foundations, roofs, facades, and windows), andfor other annex parts of the household, gates and fences included. The visual appearance of houses regarding their fitting into the rural and natural landscape, as well as regarding their placing within courtyards, are also considered and explained. The architectural guide for Ţara Oaşului bans the import of architectural elements and ornaments from other cultures, clearly stressing the fact that they powerfully contrast the local culture, for example, as seen with the pastiches and sometimes the ridiculous and obsolete imitations of buildings copied from other regions of Europe. The Romanian legislation was recently improved to include a series of measures for conserving the aesthetic/architectural features of the built-up framework while also preserving the natural landscape, aiming to respect and capitalise on the specificity of local heritage. This indicates that Romania has overdue already sanctioned urban plans and related local regulations, which govern the structural and architectural rehabilitation of traditionally constructed buildings [34]. The recommendations expressed in the recently published architectural guides are not yet included in the administrative legislation at the regional and local level in Romania, and the settlements in Ţara Oaşului are no exception.

*1.2. Transformations of Traditional Architecture in Rural Settlements of Romania*

Traditional Romanian architecture evolved gradually, adapting to people's needs but always reflecting the environmental, cultural, technological, economic, and historic living conditions of each region [35–38]. As such, old houses constitute living documents about our forebears and make them a valuable tool for learning about the development of each region's material culture. They are distinguished through their small sizes, simple construction methods, balance of their proportions, and harmony of their decorative ensemble. In the past, they were built from local natural materials according to plans known by the village craftsmen, matching their landscapes and adapting to their environment [37,39–41]. This type of buildings dominated the country's rural areas until the first part of the 20th century.

The first wave of deterioration of the rural built-up environment started in 1950 and amplified between 1970 and 1990 when the old construction materials (wood and stone) were replaced by bricks and autoclaved aerated concrete [37,42,43]. Additionally, the

communist regime began aggressive and even utopian urban policies in the 1970s. It accelerated their implementation during the next three decades, which irreversibly modified the landscape of some rural Romanian localities by partially or fully urbanising them [44,45]. A measure of the scale of these urbanisation efforts is the 1989 Villages Roumains Operation protest movement, which sensitised public opinion and generated both mobilisation and support for the Romanian villages on behalf of local Belgian communities [46].

Another wave of changes to the traditional architecture of numerous Romanian rural areas occurred after 1990. The leading causes included: opening of the country's borders, an influx of foreign investments mainly in the western and central parts of the country, diversification of available building materials, and even the preference of inhabitants towards oversized constructed buildings [47]. This is the period of "maison d'orgueil", which invaded Romanian villages and are recognisable by their aspect of urban villas embodying a combination of styles and sharing one primary feature: excess (of rooms, ornaments, and size) [48]. Extreme examples of such buildings are the opulent Roma palaces. This phenomenon resulted from the import of international architectural styles and the legislative voids of that time [43,49].

After the anti-communist revolution of 1989 [50], there were efforts at a national level to restructure the country's economy through privatisation, but many failed, which led to the collapse of the labour market and an increase in unemployment [51]. As a result, after Romania's accession to the EU in 2007, approximately 2 million people went abroad in search of better working conditions [52]. These out-migration flows were responsible for the gradual insertion of western European architectural styles in Romania. Thus, the last 30 years of experiencing western models have distorted the essence of local customs and traditions previously rooted in the Romanian psyche. The recent transformations of the traditional Romanian village as a result of massive migration abroad are unmistakable. Furthermore, the import of construction designs from EU countries modified the local traditional architectural patterns considerably, which means that in some regions of the country and in the study area especially, we are addressing a "borrowed" landscape. Simply taking construction designs from other parts of the world without considering their integration in the local pattern is a sure way towards kitsch and the destruction of the local specificity [37]. Analysing this process is helpful for housing policies or sustainable economic development strategies for Romanian rural regions, which possess a remarkable authentic cultural heritage and should be seen as an essential resource worth maintaining and promoting. In the study area particularly, the houses are even larger and more exaggerated, with the kitsch factor going towards extreme and even ridiculous levels, making it necessary to study the causes and to explore populations' perception of the final product.

*1.3. The Particularities of Țara Oașului' Traditional Architecture and Its Transformations in the Background of Its Socio-Economic Features, Migration, and Tourism Flows*

The Romanian village's main tourism advantage is its authenticity; as such, maintaining regional specificity becomes necessary in the context of globalisation. Such an example, from a physical-geographical perspective, are the villages located in the Oaș Depression and, from an ethnocultural perspective, the villages in the Țara Oașului region [53–55]. From a cultural point of view, this stand-alone region is predominantly rural, with 33 villages grouped into 11 communes and a single town (Negreşti-Oaş) [56]. The oaș human community is a clearly defined one whose people identify with the area they occupy but who nowadays face a contradiction. On the one hand, inhabitants are keen to keep their rituals, customs, and traditions (e.g., marriage, birth, funeral, music—țâpuritura, dance, folk costumes, and crafts), while on the other hand, the traditional architecture is continuously changing due both to globalisation and the younger generations' preference towards a more modern lifestyle [42,57–64].

When the country opened its borders in 1990, the oaș people were the first to leave and work abroad in France, Italy, Spain, Portugal, Belgium, Netherlands, or Great Britain. The study area has seen high remittance flows due to international migration (as people

working abroad would send part of their revenues to those at home) [65]. Most investments went towards constructing new houses or improving the already existing ones. This type of behaviour was generalised for other countries in East-Europe as well (i.e., Republic of North Macedonia) where, as a result of massive emigration, the traditional rural village had seen significant transformations [66].

The study area is Certeze village, the settlement of the Ţara Oaşului region most affected by the modernisation process. It is located in the north-western part of Romania and is included in the commune with the same name (Certeze), along with two other villages, namely Huta Certeze and Moişeni [55,67] (Figure 1).

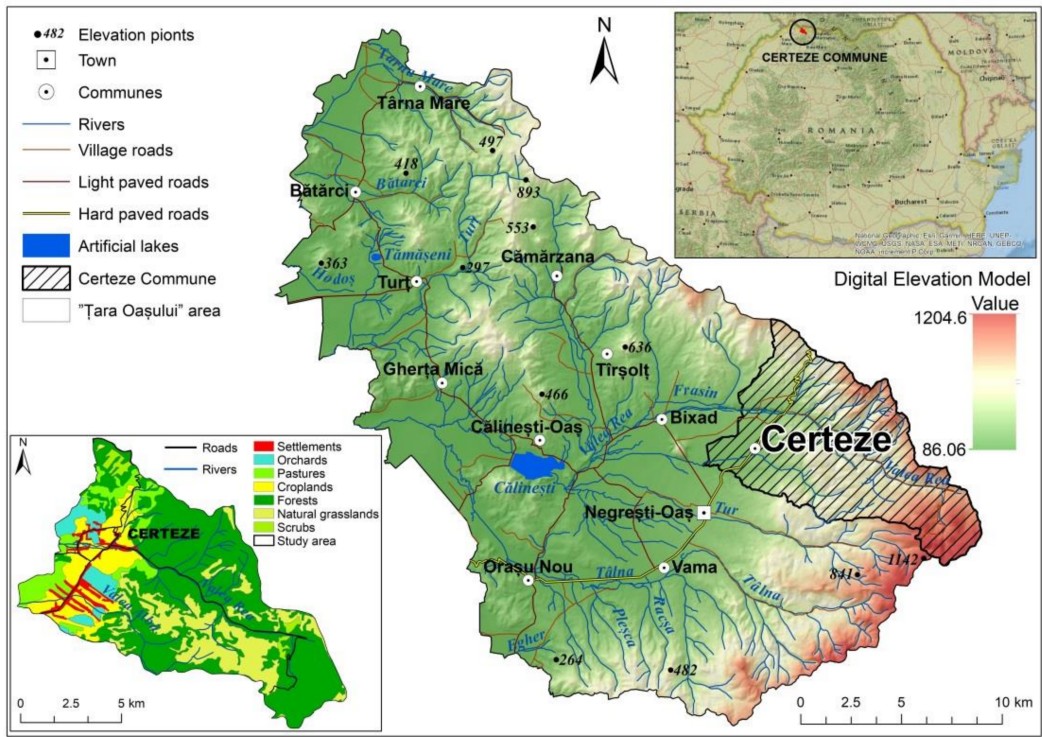

**Figure 1.** Geographical description of the study area.

Between 1990 and 2011, the out-migration flows were more substantial, fuelled by the same factors affecting the entire country [51] (Figure 2). According to the last census, the village had 3229 inhabitants in 2011 [68]. Currently, approximately 80% of people living in the village work abroad and usually have a primary or high school-level education; the remaining 20% work within their local community or within other regions of Romania [67]. The gender structure of the emigrating workforce shows that men usually work in construction and women work in housekeeping or look after the elderly and/or children. This provides them the opportunity to have direct contact with and acquire knowledge of internationally more prosperous house designs and living conditions. Considering that most of the active population works outside the village, the local rural economy is not very diversified and is based on agricultural activities, animal husbandry, and fruit tree growing [69]. The entrepreneurial sector in the area is dominated by construction or other services associated with its rurality, such as agritourism. Half of the local economic agents are active in construction and interior design, sustained by the impressive pace of the construction of approximately 25–30 new houses/year [70]. The remaining industrial activities cover the exploitation of construction materials, mineral waters, and wood.

Certeze is also the starting point of a series of tourism trekking trails towards the Oaş and Gutâi Mountains [65,72]. The village is located on DN19, the national road linking Satu Mare and Sighetul Marmaţiei, and represents one of the main gateways linking the rest of the country with the highly attractive tourism area of Maramureş [42,55]. The Master

Plan for the development of Romanian tourism has identified Maramureș as one of the six areas with the most considerable potential for tourism development at the national level [73]. Furthermore, the Maramureș region is explicitly promoted for its authentic values, specifically for its traditional rural architecture and customs, and its leading tourist attractions are the wooden churches included in the UNESCO heritage list [74].

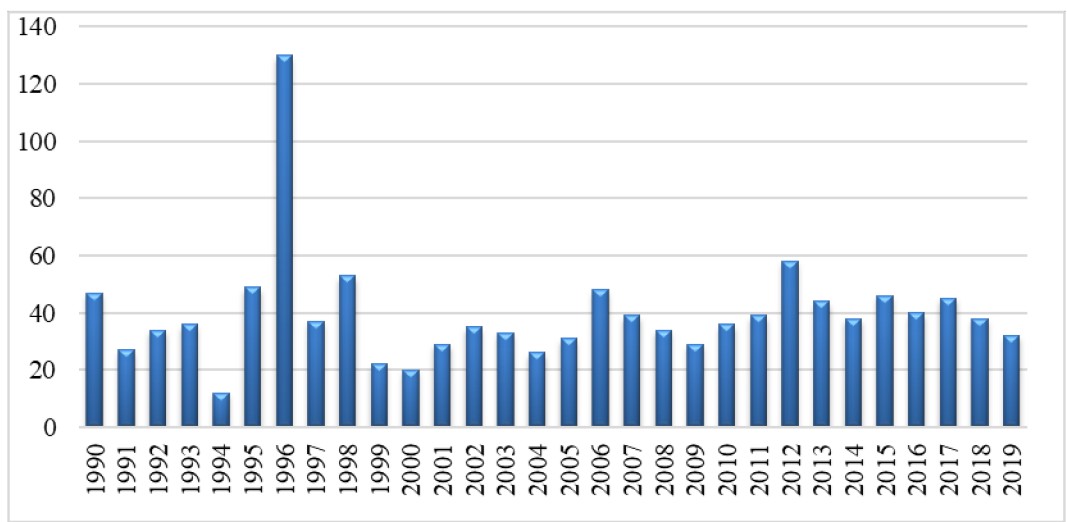

**Figure 2.** The number of departures from Certeze* between 1990 and 2019. Source: NIS, 2021a [71].

Consequently, tourism represents an essential domain for the Țara Oașului and Certeze villages. It represents a development factor because it encourages the diversification of local economies of predominantly agricultural rural areas and motivates sustainability by maintaining traditional architecture and lifestyle. The National Institute of Statistics (NIS) data shows an increase in tourism flows for Certeze from 820 arrivals in 2004 to 5000 arrivals in 2007, followed by a constant decrease until 2019 with 800 arrivals (the last reference year before the COVID-19 crisis) [75]. This proves that the area registered important tourism flows in the last decade and has real potential for attracting tourists in the future. However, attractive products and services should now address the impressive number of visitors not counted by statistics who just pass through the village on road trips often aimed at other overnight destinations.

## 2. Materials and Methods

The aim of the study is to analyse traditional rural architecture as it faces the pressure of post-modern construction standards, threatening the landscapes' authenticity and aesthetics.

The objectives of the study are as follows: (O1) identifying and explaining the causes of the alterations observed in the ethnocultural specificity of the traditional architecture in the study area, and (O2) evaluating the perception of the endogenous (locals) and exogenous (foreign and Romanian tourists) populations regarding the challenge of preserving vs. modernising traditional architecture, with the latter trend being very prevalent in Certeze village.

### 2.1. Identifying and Mapping Traditional Houses through Architectural Surveys, Topographic Maps, Spatial Images, GIS Techniques, and Photographs

The research team first went into the field between 15 and 20 May 2019 to complete an architectural survey through field observations sheets that examined the aesthetic and architectural characteristics of the houses/households in the village of Certeze, specifically the mixture of traditional and post-modern elements, and the insertion of non-native elements. Architectural surveys were successfully used to understand buildings' construction



stages, materials, and techniques, and are valuable for studies interested in inventorying and mapping houses with traditional architecture [76,77]. Furthermore, field surveys, spatial mapping, and inventorying of houses through GIS and photo techniques have also been used by scientific studies interested in ancient vernacular residences of traditional villages [14].

The other stages of the methodology required applying GIS and remote sensing techniques; processing aerial/satellite images; collecting statistical, vectorial, and cartographical data; and constructing databases, all in order to identify the field distribution of houses built before and after 1990 in Certeze village. The landmark year chosen was 1990 because the time after 1990 saw the most changes in terms of building typologies and also because 1990 marks the year the country transformed considerably from a political and economic perspective, which had a snowball effect in numerous other domains.

For identifying the houses built before 1990, the authors used the 1989 declassified topographic map sheets made by the Military Topographic Directorate at a 1:50,000 scale [78]. These were obtained in .jpg format at a 300 DPI resolution and georeferenced in the Stereo 70 projection. The vectorial data obtained from this cartographic information, specifically the old constructed buildings from the village, were later validated with Corine Land Cover data from 1990 on Romania acquired from the database of the European Environment Agency (2019) [79]. To identify the houses built after 1990, the authors used Landsat satellite images from Google Earth Pro with a 50 m resolution from 20 October 2020. These images helped to identify the newly constructed buildings from Certeze village. This information was validated using Corine Land Cover data from 2020 on Romania from the European Environment Agency (2020) [80] database. The cartographic material for the study was finalised using Arc Gis Pro 2.2 and Google Earth Pro.

Similar to other studies [14,77], apart from field observations, the authors photographed the inventoried houses (traditional, mixed, and post-modern) in order to recheck and validate the results of the classification done through the in situ observation sheets before localising and mapping them. Through the field observation sheets, this research study aimed at producing an inventory of the houses in Certeze village. They were further classified according to their construction period and architectural style, outlined through specific elements identified by the authors during study field trips (Table 1). In this way the following three categories were identified: vernacular houses displaying specific traditional architectural elements and built before 1950; houses with a mixed architecture built between 1950 and 1990; and houses displaying a post-modern architecture and built after 1990. Defining a mixed architectural style as a distinct type among the three seems perfectly justified. The numerous houses belonging to the second architectural type obviously selected elements from a vernacular style (wooden doors and windows) used particularily before 1950 and combined them with modern updated construction materials (e.g., bricks, cement, tiles, tin, etc.), corresponding to the period between 1950 and 1990.

### 2.2. Endogenous vs. Exogenous Perception: Sampling Target Groups, Interview Surveys, and Data Visualisation

The second round of field visits was conducted between July 28th and August 10th, 2019. The interval overlaps with the summer holiday season to ensure that the authors were able to interview both migrant workers who come home for the summer and tourists, as July–August represents the peak tourism season of the year. In order to evaluate the endogenous and exogenous perception of the local architecture, the authors conducted 60 semi-structured interviews: half with the locals (endogenous perception) and the other half with foreign and Romanian tourists (exogenous perception), most of whom were transiting Certeze village towards other destinations, particularly Maramureș. The tourists constituted the exogenous perception because they are not active decision-making factors for the local architecture as opposed to the locals who implicitly form the endogenous perception. All the interviews were realised face to face and then both manually processed and coded to maintain the respondents' anonymity. The respondents agreed to answer

all questions and allowed for the information to be used for scientific research. When addressing foreign tourists, the interviews were conducted in English.

**Table 1.** The architectural styles present in Certeze village and their specific elements.

| Architectural Style | Period | Criteria | |
|---|---|---|---|
| Vernacular architecture | Before 1950 | ✔ | shingle roof |
| | | ✔ | wooden porch |
| | | ✔ | clay and/or wooden walls |
| | | ✔ | wooden windows |
| | | ✔ | wooden doors |
| | | ✔ | natural stone foundation |
| Mixed architecture | Between 1950 and 1990 | ✔ | tile or tin roof |
| | | ✔ | brick walls |
| | | ✔ | walls plastered with cement |
| | | ✔ | wooden windows |
| | | ✔ | wooden doors |
| Post-modern architecture | After 1990 | ✔ | tile or tin roof |
| | | ✔ | brick or concrete walls |
| | | ✔ | wooden or crick loft |
| | | ✔ | modern PVC or stained-glass windows |
| | | ✔ | carved wood or PVC doors |
| | | ✔ | decorative allochtone elements on the facades (made out of wrought iron, stainless steel, concrete pillars, etc.) |

Since our study was qualitative, respondents were identified through non-probability sampling techniques [81]. In addition, the sampling was stratified for both target groups as the interviews aimed to cover all age intervals (young people aged between 18 and 35; adults aged between 36 and 60; and seniors older than 60 years). More than half (53.34%) of the locals interviewed were seniors, followed by adults (36.66%) and young people (10%). Conversely, the tourists interviewed were mainly adults (60%), with the remaining evenly split between seniors and young people (20% each).

In terms of economic structure, the locals covered the following professions: local administration, professors, priests, local entrepreneurs, workers (many active in the study area but most working abroad), and pensioners. Workers composed 80% of this target group and had either a primary education (13.34%) or had graduated high school (66.66%). All the interviews took place inside the house or household of each respondent.

The second target group was composed of one-day and overnight visitors, both foreign and Romanians, who covered, in terms of occupations, a spectrum broader than the locals (i.e., architects, engineers, IT specialists, doctors, entrepreneurs, and workers). Most tourists had superior levels of study (80%), with the remaining 20% having graduated high school, which confirmed the dominant profile of architectural/cultural heritage tourists with an above-average educational level [73,82]. Many of them are one day-visitors who transit Certeze village from or towards Maramureş, a region of great interest for cultural tourism circuits. About three quarters were Romanian (76.66%) living in major cities of the country (Bucharest, Târgoviște, Craiova, Sibiu, Timișoara, Oradea, Iași, and Constanța), with the remaining 23.34% derived from Germany, Israel, France, and the Czech Republic.

This study used an in-depth qualitative approach, namely interviews and detailed narratives for both target groups, which are essential for understanding a personal point of view and a micro perspective regarding a continuously changing and complex social phenomenon [81].

In order to establish the endogenous perspective, the interview guide contained open questions on respondents' profiles (age, gender, education, and profession); the location and comfort level of their dwelling (the year the house(s) they own was/were built; the location of their house(s) inside their property; motivation for building a newer house

on the same parcel as the old one; the practicability level of the new house in relation to its size; whether the new house is used as a dwelling or not; and a review of the comfort elements of the newer house); the aesthetics of the construction (how the design of the newer house was chosen and whether they resorted to specialised advice in choosing the architectural style or not); and the significance of modernally constructed buildings inside the community (what is the message they transmit to or receive from their neighbours through the large buildings and a one-word description of the houses in their community).

In order to establish the exogenous perspective, the guide contained open questions on respondents' profiles (age, gender, education, and profession); country of origin; reason for being in Certeze; source of information about the village, if any; the manner in which they see the newer houses compared to the older ones; opinion on the mixture of architectural styles; and opinion on the size of the newer house.

Additionally, the data obtained from these interviews were processed using Word-Clouds.com to create a visual representation of the tourists' opinions regarding the architecture of the houses in the village. This method is relevant in perception studies (including tourism), as it offers a comprehensive image of the frequency and, as such, the importance of the words used by the respondents through their size and placement in the word cloud.

### 3. Results

#### 3.1. Identifying and Explaining the Alteration of the Traditional Architecture Specificity

According to the literature on the topic and the architectural survey, a traditional household in Țara Oaşului is surrounded by a twig fence and contains a wooden beam house built on stone foundations with a wooden porch. The house is usually not painted and preserves the natural colours of the traditional building materials; its windows have wooden frames and sometimes wooden shutters, and its roof is made out of straw, shingles, or ceramic tiles. Locals usually carve symbols in the frames of the doors and windows, such as ropes, rosettes, or rhombuses. The same geometrical symbols could also be found on the horizontal or vertical boards of the porch. The household usually contains a multifunctional barn (cattle stable, summer kitchen, and other functions) and small-sized outbuildings [83] (Figure 3). Romania is defined fundamentally as a civilisation of wooden—regardless of the essence—buildings; wood was a traditional reference construction material that defined the Romanian rural landscape [48].

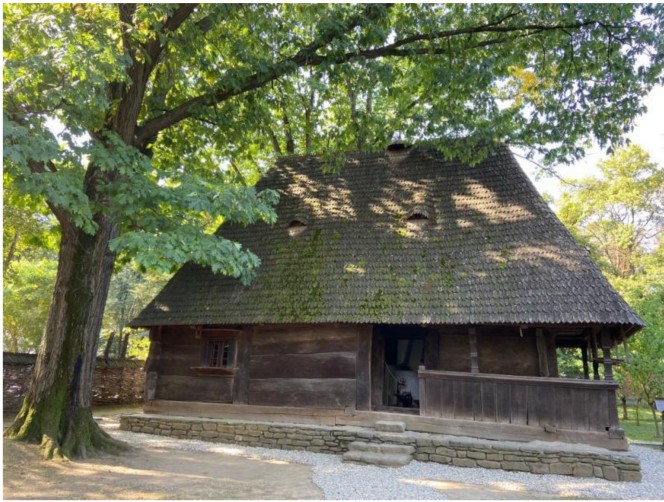

**Figure 3.** Traditional house with stone foundation located in Moişeni village in the Certeze area. Photo: A. Taloş, 2021, at the National Village Museum "Dimitrie Gusti", Bucharest.

Based on field research, architectural surveys, observation sheets, and GIS technologies, the authors detected three types of buildings according to their construction period and architectural style (Table 1).

The oldest architectural style overlapping the genuine traditional one refers to vernacular houses. They were built before 1950 from local materials (wood, stone, and clay), displaying a peasant house specific porch, with two or maximum three rooms of a modest size. Research trips revealed only four houses of this type currently existing in Certeze village, from which one of them hosts the etnographic museum.

The second architectural style is the mixed one, dated between 1950 and 1990, when new houses replaced the vernacular ones. They were much bigger than old houses, displayed an upper floor, and used updated construction materials for the brick walls and tile or tin roofs. As they used traditional elements, these houses still kept wooden doors and windows, which are reminiscent of peasant architecture.

The most recent architectural style refers to the post-modern one. It emerged after 1990 and is characterised by imposing, large-size houses displaying contrasting elements in terms of both dimensions and colours. They were mostly built from concrete and brick; have one, two, or even three levels; wooden or brick lofts; tile or tin roofs; modern PVC and sometimes stained-glass windows; carved wood or PVC doors; and balconies incorporating elements unusual to the area (e.g., wrought iron, stainless steel, and concrete pillars) and are surrounded by concrete or wrought iron fencing. Each house has a contrasting colour, architectural design, and they became veritable kitsch elements inserted in a landscape with significantly different cultural values. During the field research, a total number of 1.869 houses were mapped. As houses built before 1990 display vernacular elements in their architecture, the authors considered this period influenced by traditional. In total, 924 houses (49.44%) built before 1990 were identified, from which only four were built before 1950. The remaining 945 houses (50.56%) are new buildings etablished after 1990 in a post-modern style not specific to Ţara Oaşului (Figure 4).

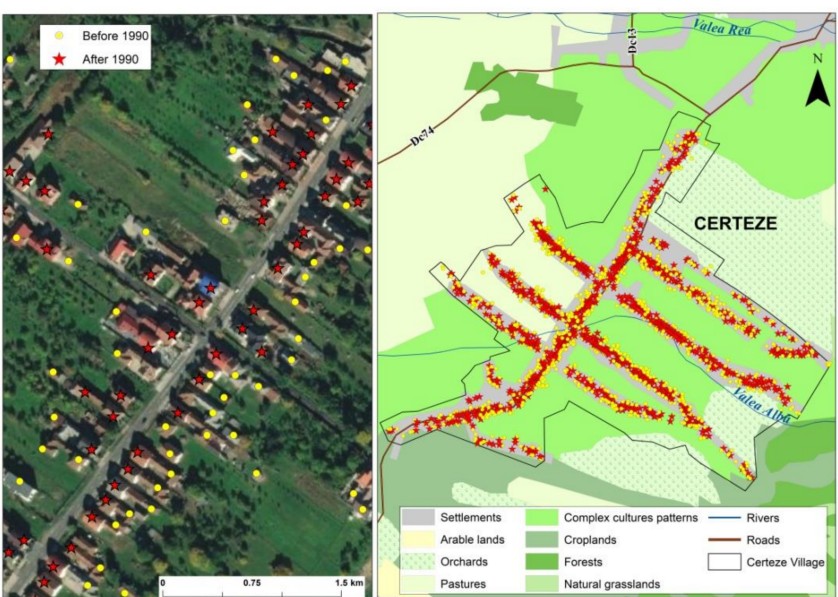

**Figure 4.** Distribution of houses built before and after 1990 in Certeze village.

Some old houses are located between the newer ones, usually behind or next to them (Figures 5 and 6). According to the data obtained from mapping, more than three quarters of the old houses are on the same land parcel as the newer ones. In the study area, older houses (12.58%) overtake newer ones (8.45%) concerningt those that solely occupy one parcel. Most of the old houses that occupy an entire land parcel are located towards the fringe of the village, far away from the main road.

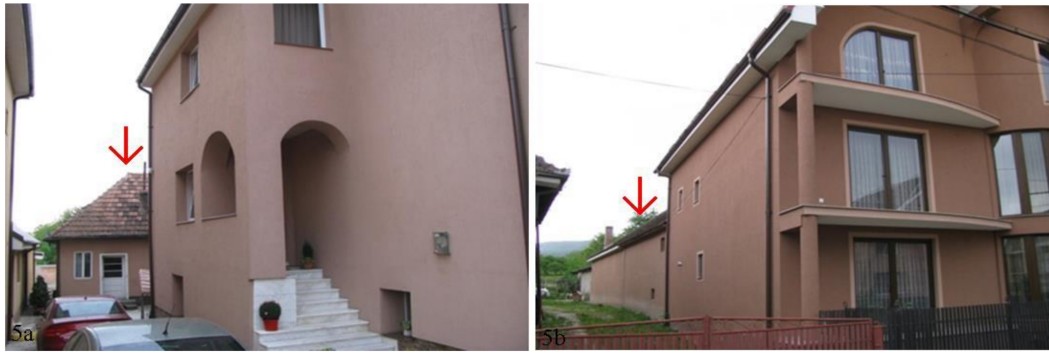

**Figure 5.** Traditional houses placed behind or right next to the newer ones. Photos: M. Preda, 2019.

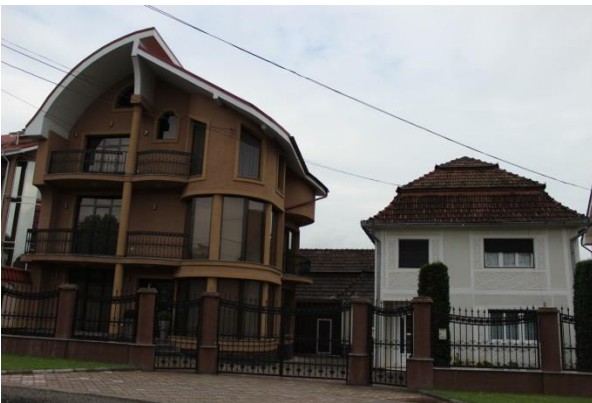

**Figure 6.** An antithesis between old and new in Certeze village. Traditional vs. post-modern buildings. I. Vijulie, 2019.

In addition to their peripheral location, many of the traditionally built houses also find themselves in different stages of deterioration, even though they are genuine treasures as well as living historical testimonies of locals' skill and diligence, representing their efforts to pass on Romanian traditional architectural elements. Furthermore, because these houses are not included in any rehabilitation project, one of the connections between locals and their history will be irremediably lost if and when they fall or are destroyed.

There are, however, examples of good practice: one house dated back to 1947 in the centre of Certeze village was transformed into a museum and is visited by those on road trips in the region (Figure 7). This kind of endeavour should continue alongside efforts to save other traditional buildings; nonetheless, this would require a more active involvement from local authorities.

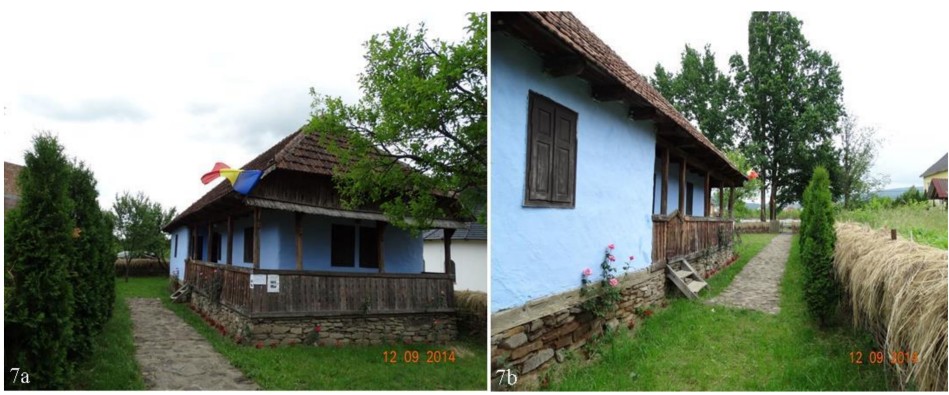

**Figure 7.** Museum house in Certeze village. Photos: A. Corzan, 2014.

The discussion with the museum curator confirmed that even the older houses, built before 1990, were modernised during the post-communist years in order to resemble their counterparts from more developed countries. Furthermore, it was only until 1950 that the houses in the village kept a traditional typology: the lines were simple and the construction materials were those readily available: wood, stone, and clay. After that point, construction started to become evermore imposing, meant to mirror the financial power of the owners (woman museum curator, 61 years old, Certeze).

Data extracted from the observation sheets showed that the newer buildings did not fit with the traditional elements specific to the region (Figure 8).

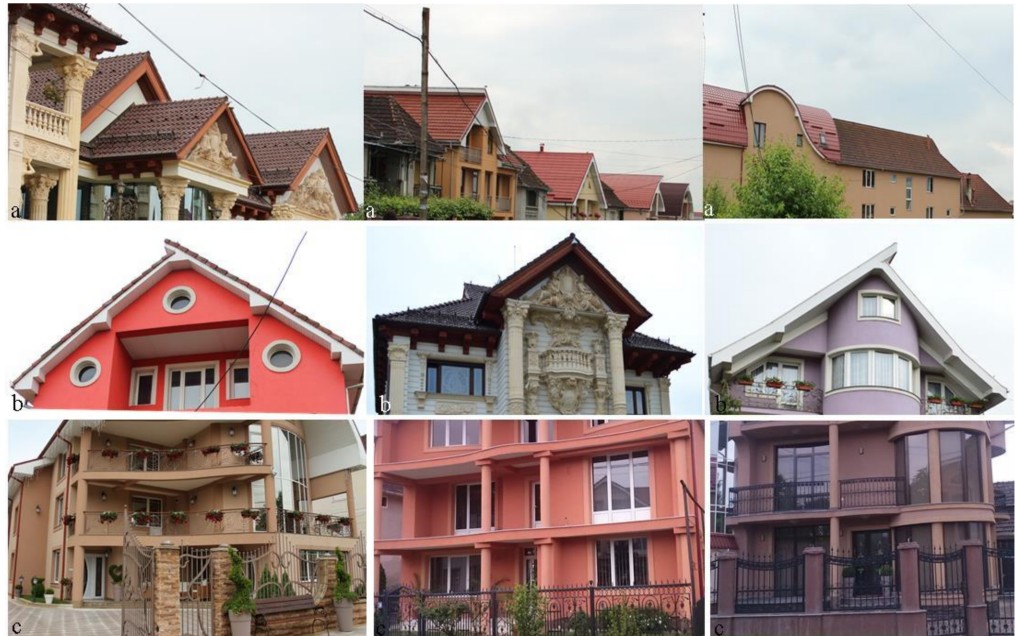

**Figure 8.** Elements of post-modern architecture in Certeze village: (**a**) roofs, (**b**) gables, and (**c**) facades. Photos: I. Vijulie, 2019.

The contradiction between current urban planning regulations and the real ground situation is visible everywhere in Certeze village. The representatives of local authorities declared that they had imposed the measurements of the General and Local Regulation of Urbanism as per the methodological norms [84] regarding the authorisation of construction works, which were republished with further modifications and addendums in 2001 [85]. In the Romanian legislature, after any building is finished, it must undergo a "reception phase" in order for it to be connected to the sewage or electricity network, this being the last stage in which any inconsistencies of any kind can be identified. However, no sanctions were applied in Certeze in the reception phase for those who did not comply with the law and constructed without a building permit or violated the permit's provisions. Thus, in most cases, the construction works continued, no measures were taken to ensure that the buildings would conform to the permit's provisions, and no building was demolished. Local authorities declared that they took a series of more decisive actions starting in 2019. They approved the establishment, within City Hall, of a department specialised in the field of urbanism, landscaping, and building permits, with demolishing decisions power, which, following field visits, would have the authority to take action against those who did not comply with existing construction regulations [86]. Nevertheless, this decision contradicts the opinions posted on the City Hall website that promotes and encourages the post-modern architecture of the village as well as the new aesthetic that is slowly finding its place in the collective subconscious of the village:

> *"once you reach this locality, you will be amazed by the beauty of the place, the modern houses . . . . Each house has a different colour, and each household tries to show you*

> *something new. At the same time, you have to see how clean and tidy the houses of the locals are. Everything you see proves how hardworking local people are"* [87].

### 3.2. Evaluating the Endogenous and Exogenous Perception regarding the Contrast of Post-Modern vs. Traditional Architecture

3.2.1. Locals' Perception on Preserving Traditional Elements vs. Modernisation

The data obtained through semi-structured interviews proved that locals' perceptions of traditional vs. post-modern architecture differ according to their age. The older respondents preferred the traditional architecture unanimously, while the young and adults segments opted in their majority (92.85%) for the more modern houses built from state-of-the-art construction materials of generous dimensions, far exceeding the needs of a standard-sized family.

The population that migrates to work only returns home on legal (e.g., Christmas, News Year's Eve, Easter) or summer holidays. For the rest of the year, the village is inhabited mainly by the elderly and the very young who do not live in the new buildings. The older participants in the survey declared that the house they are actually living in— usually a modest two or three-room old house—is placed on the same plot next to the more modern buildings. This also confirms the conclusions of the observation sheets and GIS measurements. When explaining why the newer—five to seven rooms—houses were located on the same plot as the older ones, on the land their family had inherited from previous generations, the interviewed adults declared that this was a tactic used because the available plots facing the main road were limited and new plots available for construction are expensive. According to the interviewees' answers, the older family members never become familiarised with the newly constructed buildings that the younger, working, and living abroad members build. Therefore, these newer houses continue to remain empty, waiting for their rightful owners.

> *"My wife and I live together with our grandchildren in our old house; it was built in 1957"* (pensioner, male, 84 years old).

> *"I received my house from my parents . . . as long as I live nobody will demolish it because I will not allow it . . . one of my boys has built this big house in front of it"* (pensioner, male, 76 years old).

> *"I have a lot of beautiful memories from my youth connecting me to my old house, so I still live in it; I did not want to have it demolished, even if my children built this villa facing the street and they are very proud of it"* (pensioner, woman, 72 years old).

> *"Nobody lives in the new house; we work in France"* (male, construction worker, 57 years old).

Some interviewees declared that the new houses are uninhabited because even the younger generations sometimes live in the old houses.

> *"When we come back home, we still live in our parents' houses, the new ones, we keep clean"* (woman, housekeeper in Italy, 32 years old).

> *"Our houses are clean, ready to receive guests; we keep them that way to boast"* (woman, housekeeper in Spain, 52 years old).

The respondents also emphasised that large buildings are not necessarily something new for their village; people have always built big, even before 1990, and this trend only continued. At first, after 1990, people who worked abroad used that money to modernise their houses and only after that they started constructing new houses that would resemble the western European ones. Our fieldwork emphasised that the oversizing of the new constructed buildings sometimes takes the form of a ridiculous competition, marking an acute need to astound and gain social esteem.

> *"My neighbour to my left built around 1997 a 1-floor house and after two years my other neighbour built a 2-floors house . . . what do you think the first neighbour did? He took*

*his roof down and added another floor to his house. So now both houses have the same
height . . . here nobody outdoes anybody..."* (pensioner, male, 80 years old).

When questioned on how they chose the aesthetic of their future house, respondents
declared they recevied inspiration from what they saw in the more developed countries in
which many of them worked in construction (France, Italy, and Spain). They acknowledged
not using specialised counselling in deciding the architectural style and emphasised that
they used the money they worked hard to earn in order to build their house just the way
they wanted to.

Their pride and sense of achievement surfaced again when asked to give a short
description of the houses in their community. The descriptions and comparisons include:

*"beautiful houses; proud houses; colourful houses; brilliant houses; diamonds; extraor-
dinary dwellings; tall buildings; mind-boggling constructions; constructions like in
western Europe; houses like nowhere else; the most beautiful houses in the country".*

### 3.2.2. Tourists' Perception on Preserving Traditional Elements vs. Modernisation

According to the word cloud that resulted from tourists' answers (Figure 9), the
most often encountered words and phrases were traditions, old architecture, preservation,
cultural identity, and tourism, demonstrating that traditional architecture is a significant
motivating factor for tourism in this area. Tourists, mostly urban dwellers, were interested
in finding the traditional facets of the rural area they visit, mainly because this daily lifestyle
has disappeared in the area they originate from. The opinions of the interviewed tourists
tilted unanimously towards the need to preserve the traditional architectural elements of
the built-up heritage, with 93.4% disagreeing with the modernist manner of constructing
the facades of the houses and only 6.6% declaring that they consider the present houses in
the Certeze village beautiful.

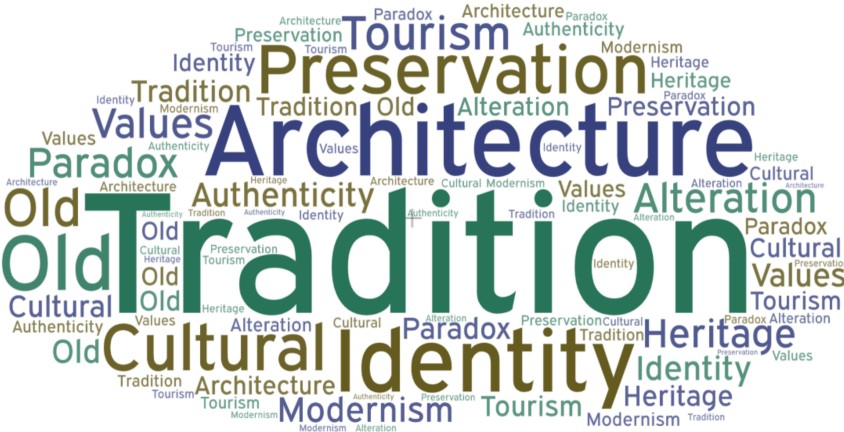

**Figure 9.** Tourists' perception of the Certeze village houses' architecture. Source: interview guide
data processed by the authors.

Most tourists had a topo-rejection attitude towards the newer constructed buildings,
seeing them as contrasting, paradoxical, and responsible for distorting the traditional cul-
ture of this rural area as well as its authentic architecture. At the same time, they expressed
their preference for the traditional house from this ethnocultural region. They agreed that
authorities should take action to stop the destruction of the authentic architecture and
also to save the remaining traditional buildings in the village. Tourists mentioned that
the architecture of the buildings needs to harmonise with the scenery and fuse with the
natural landscape.

*"I think we should continue to show compassion to the old houses because they are
history's witnesses in each village, and they should not just disappear"* (woman, 34
years old, Târgoviște).

*"I can see a real paradox here in Certeze: the locals are renouncing the traditional elements and transform the very things that tourists would like to see—the authentic landscape of this place. I think they want to have modern living conditions, but in the process, they irremediably lose the values that we, tourists, seek"* (woman, 28 years old, Craiova).

*"I can see an unusual proportion of the construction on one plot; the buildings are oversized, they seem to have a role of representativeness for the owners rather than housing"* (male, 42 years old, Bucharest).

Even if most people traversing Certeze describe its architectural design as unattractive, it nonetheless arouses their curiosity. Tourists' interest in Cereteze stems from its famed, reinterpreted eclectic style, oversized buildings that amaze and surprise, and its wow effect, which is sought after by every tourism destination [88]. In total, 86,66% of the interviewed tourists declared knowing about Certeze village from mass media, where it has been the subject of news for years [89–93]. Respondents (93.33%) also confirmed that they chose to transit the village because of its fame, even though many criticised the post-modern architecture and continued their journey towards the more established tourism areas. Only 6.67% declared that they did not plan to be in the village.

The results confirm a twofold theory: the role of mass media in proliferating tourism sites and the globalisation that takes place under the "tourist gaze" in its many forms and manifestations [94,95].

## 4. Discussion

Traditional architecture is one of the main tourism-attracting elements in the area, with its value being emphasised both by Romanian and foreign tourists. Nevertheless, due to a series of factors, Certeze village has moved architecturally towards a post-modern style with inhomogeneous, contrasting trends to the detriment of the pursuit of conserving traditional heritage elements (Figure 10).

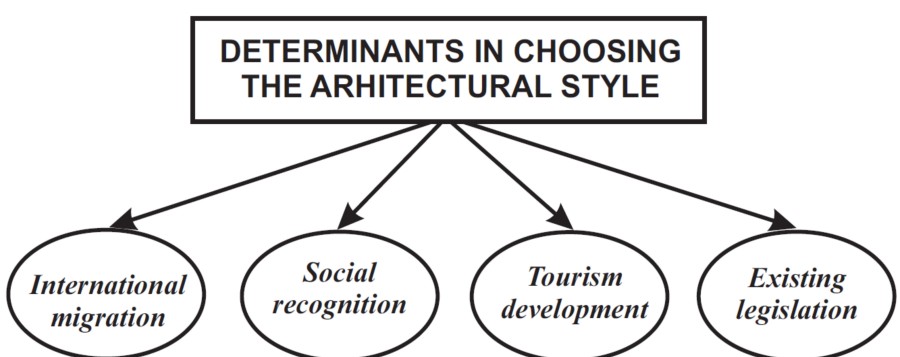

**Figure 10.** Determinants in choosing the architectural style.

The international migration of the labour force is one factor determining the radical architectural changes. This factor increased incomes for the local population while also presenting them with the available endowments for improving quality of living, which overlapped auspiciously with the emancipation of the younger generation who no longer desired to be involved in agricultural activities. Among inhabitants, this created the need to increase living comfort, which is to the detriment of keeping traditional architecture. As such, many locals who migrated towards western Europe used their earnings to modernise their houses or build entirely new ones. Investing their money primarily in housing could be explained by the lack of entrepreneurial actions of the untrained ex-agricultural workers and by the limited opportunities for investing, apart from housing, immediately after 1990.

A second factor relates to social recognition. For the locals, the house represents a business card, an emancipation symbol, a token that demonstrates the owner's economic and social status, and a form of accumulating fortune, which is also easy to display to community members. When referring to the neighbouring region of Maramureş, one

study [96] interpreted the prominently adorned houses as a need for social recognition and an exacerbated feeling of self-esteem, which manifested in the case of Țara Oașului through an obvious oversizing.

A third factor is the potential for tourism development in the village, with tourists pressing the local community to conserve their ethnographic heritage, including architecture. According to official statistics, overnights in the study area are low. Nevertheless, the village is visibly transited by an impressive number of tourists, heading mainly to Maramureș, the neighbouring cultural area attracting both Romanian and international visitors. Regardless of the length of stay, visitors' presence could and should be used to promote the existing architectural heritage and signal its slow but inevitable disappearance. A 2015 study [18] underlined that new architecture should not imitate the old one but should respect the local specificity to sustainably develop the area in the future and attract tourists interested in the authenticity of rural regions. The youth and adults of Certeze do not currently use the luxurious villas for housing; they see them rather as a statement of social recognition and honour for their family, and also as a possible accommodation opportunity for guests or tourists. One last important factor was the legislative voids in place at the moment of construction. For a long time, there were no explicit, compulsory norms to govern buildings' typology in areas that are recognised for their ethnographical value. This resulted in contrasting residential landscapes.

Another specialised study [97] found that post-modernism changed the faces of houses in Certeze while the traditional became "outdated". The insertion of modern elements in domestic daily life, the need for social recognition and the competition between villagers at a local level, the legislative void, and the lack of authorities' involvement in the conservation of the architectural heritage led to the birth of atypical landscapes, with obvious consequences regarding the aesthetic and authenticity of the rural cultural environment not only in the village of Certeze from Țara Oașului but also in other areas such as Maramureș [48].

The social prestige is directly proportional to the dimension of the house and such competitions that gain fantastic levels are present everywhere in Țara Oașului, as well as sometimes even in Maramureș; these aspects are part of a socio-cultural pattern and a typical regional lifestyle, as also observed by other studies [48,61]. This validates other findings which concluded that the preference of main roads as a location for post-modern buildings is due to the fact that such a position ensures the house's visibility while also better proclaiming the owner's social status.

Conserving the traditional architecture in Certeze village is pivotal due to its regional context: the village is part of the Țara Oașului region, which is in itself a distinctive ethnocultural area of the country, but more importantly, is one important gateway for road trips towards the Maramureș tourism region, a brand promoted precisely for its preserved ethnographic and cultural heritage.

During the interviews, tourists, especially the foreign ones, said that when visiting this part of the country, they have specifically refused to spend the night in a "modern" house, seeking either the old traditional houses or preferring to sleep in hay barns for an authentic experience. This could be of concern to locals in the study area who are interested in developing a tourism business. Therefore, the present research emphasises the need to inform locals in order to better acknowledge the potential value of conserving old traditional houses and stopping the ample process of destroying the traditional architecture. Although it seems improbable that the "transformation through preservation" method adapted to the 21st century will be possible here, tourists' insistence regarding the importance of conserving the local specificity could represent a warning signal, at least for those locals that pursue tourism entrepreneurship or foresee a possible occupation in tourism.

There is, however, a small percentage of tourists who appreciate the new construction trends and consider the new buildings as beautiful. While currently the traditional architecture has dwindled and been replaced by a somewhat homogenous landscape, dominated

by distinct and contrasting villas, nonetheless, the village attracts important visitor flows as a result of its notoriety. The current landscape could evolve in a twofold manner: either aiming to conserve traditional architectural principles or continue the present construction trend. If post-modernity is to prevail in the future and the new houses will continue to be imposing and eccentric, one solution would be their tourism capitalisation. Certeze is already famous and attracts tourists, but they do not spend the night or contribute to the local economy. Even if it is to lose its traditional architectural specificity, the settlement can capitalise on its immaterial cultural heritage, including its customs, traditions, and gastronomy, which could be promoted and sold while using an accommodation infrastructure that is atypical for the area but specific to the locality itself: villa-type houses.

The role of defining, mediating, and directing sustainable house planning, as well as the economic development of the different interest groups in Certeze village (tourism entrepreneurs, young vs. older inhabitants, villagers that currently work abroad, and villagers that work locally or in other regions of Romania, including construction projects) fall to local authorities. They, in turn, are one of the stakeholders in this equation. When approaching the "traditional vs. post-modern architecture" dilemma, one solution would be a good practices guide on preserving local specificity with clear instructions on constructing new buildings according to expertise in the field. The local administration must transpose this guide into local urban planning regulations and verify them in the field by specialised teams to ensure that its content is both respected and applied as in other similar regions of the country as well as in Europe [37,98].

Regardless of what scenario will prevail, house planning and economic development strategies are arguable topics of high interest among interest groups at the local level [99]. Certain groups of stakeholders, especially house owners, should also be financially motivated towards developing tourism activities to ensure an optimal negotiation for preserving the traditional rural architecture.

It is a reality that Certeze is losing its architectural tradition, despite its success in conservation proved through enlarged interests in traditional customs mainly related to specific holidays throughout the year. However, the village orientation toward modernism is not an absolute disadvantage precisely because its new eclectic style has gradually become its brand and can be used in updated and adapted promotional campaigns, which in the end might help maintain the existing traditions.

## 5. Conclusions

The cohabitation between traditional and modernity, with modernity weighing more, has transformed Certeze into one of the most eccentric and controversial villages in Romania, with a high risk of completely losing its traditional architecture, undoubtedly an essential part of its tangible cultural heritage. Long-held traditions passed down through generations by Certeze's inhabitants have diluted with the modernisation of the rural environment, beginning with the communist era and accentuated by Romania opening its borders towards western Europe after 1990. As a result, globalisation swept over this vulnerable community with a low to medium educational level but very skilled in construction. The traditional architectural style of the houses is mostly gone and is replaced by new, completely modern buildings. One major conclusion of this study is that it was not poverty but wealth that affected the architectural landscape specific to Certeze village, as almost all traditional buildings were demolished and replaced with oversized villas.

Older inhabitants prefer the more traditional houses while the young and adults opt for newer dwellings made out of materials that do not fit in the traditional local landscape. While tourists have the status of being a stakeholder in the attempt to preserve traditional architecture, they are not influential enough to impact the decision-making process. This is partially because there are no communication links currently being implemented between locals and tourists, and because of the significantly low economic interest of the very few local tourism entrepreneurs. The study area has experienced a lack of entrepreneurial initiatives and investments that could have diversified the local economy, which, as a result,

is still predominantly agricultural. Without positive perspectives or real opportunities to be integrated as a whole in a synergetic economy, the local labour force dispersed and found better working conditions by themselves, thus becoming fragmented and individualistic when deciding the architecture of their newly constructed homes.

The lack of legislation based on scientific principles for conserving the built cultural heritage and authorities' inability to preserve the existing heritage has exacerbated the eclectic image of Certeze village. One crucial step in this regard is creating and implementing legal frameworks for architectural regulations and participative policies meant to conserve elements considered architecturally and aesthetically valuable while simultaneously accommodating present-day living comfort requirements, alongside the harmonious development of the built-up environment. These are current responsibilities of local authorities who fail to demonstrate the necessary various and complex competencies to manage these issues.

A future study may consequently approach the local and/or regional authorities' perspectives on the complex issue of rural architecture in terms of the difficulties and rationale for preserving its traditional elements, of sustainable contemporary house planning, or of legislative gaps and current administrative practices. In addition, comparative analyses to other SEE countries, which own an important rural heritage and face powerful globalisation and modernisation influences, enhanced by important emigration flows and rising living standards would be of much interest.

**Author Contributions:** All authors have contributed equally to this research study. All authors have read and agreed to the published version of the manuscript.

**Funding:** The publication of this research study has been partially funded by the University of Bucharest, Romania.

**Institutional Review Board Statement:** The study was conducted according to the guidelines of the Declaration of Helsinki, and approved by the Ethics Committee of the Faculty of Geography, University of Bucharest.

**Informed Consent Statement:** Informed consent was obtained from all subjects involved in the study.

**Conflicts of Interest:** The authors declare no conflict of interest.

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
