# Peer review of "Certeze Village: The Dilemma of Traditional vs. Post-Modern Architecture in Țara Oașului, Romania"

_sustainability, doi:10.3390/su132011180_

Round 1

Reviewer 1 Report

This interesting case study exposes the conflictual contradictions of the prolonged modernisation of rural landscapes in remote parts of Europe. In the case in question, the specificity of the Romanian context is well-considered and stems from fieldwork and a thorough understanding of the nation's socio-economic conundrum of its inter-European migration flows. The collection of data through qualitative interviews is appropriate and has been successfully tapped to reveal the attitude of part of the local population to manifest monetary achievement by competing with their co-migrant neighbours through the construction of opulent, yet often empty, residences.
The term "modern" is used in the paper in antinomy to the traditional vernacular character of the rural village that has been analysed, and it is referred as a potential threat to the economic benefits for local tourism. While the term "modern" can mean many different things to many different people, the word does not ring true to represent the phenomenon observed by this study to an architecturally educated audience.

I suggest to change the term "modern" with the more pertinent qualifier of "post-modern." The architectural characters of the buildings observed do not conform to the modern architecture canon at all. The new residences of the Certeze village are in reality a form of contemporary vernacular that re-interprets, often with a pastiche that takes to grotesque outcomes, some forms, and motifs borrowed from historical examples, some of regional origin and some universal like the classic orders. There is plenty of literature in the field of architectural history and theory that could provide a broader vocabulary to interpret the phenomenon at stake. Further reading could provide inspiration to continue researching in the causes at the root of what has been revealed by this exciting case study.

Author Response

Dear Reviewer,

Thank you for your useful comments and suggestions we attentivelly followed for an improved version of our paper.

Please find the answers to your comments in the word document attached below and entitled:

Kind regards,

The authors

Reviewer 2 Report

Main comments on the part of the research conducted (parts 2. Materials and Methods, 3. Results) :

1) The question of inventories of existing buildings. Are there any previous studies on inventories of these buildings, both those that still exist and have been assessed, as well as regional ones that no longer exist?

If so, in my opinion, they should necessarily be included in the article (reference to these materials, bibliographic entry).

Need to organise the part concerning the inventory, assessment of existing buildings. Definition of clear criteria for evaluation of examined buildings. Clear division, grouping by time of construction. Maybe, distinguishing main regional elements and those elements, parts of the building which undergo the biggest changes. Is it possible to notice any reference to regional features in new buildings or are they mainly borrowings from foreign regions?

Maybe some list of these objects by group (supplement Figure 4. Distribution of houses built before and after 1990 in Certeze village).

Note: the article says that the field research was conducted in 2019, while the photo of the object is from 2014 (Figure 7. Museum house in Certeze village. Photo: A. Corzan, 2014.) Clarification, correction.

Generally, the authors inform what software they used, what materials they used for the inventory of buildings (location of buildings, number of buildings built). In my opinion, however, there is not much reference to the results of this inventory, the evaluation of the buildings. Obviously, the aim of the research was to examine the attitude of the inhabitants and tourists to the existing buildings and the newly constructed ones, but in my opinion, the above-mentioned issues should be discussed in more depth - as an important background for the basic research.

Tidy up also in terms of presentation of graphic material (photos of objects). The current form seems a bit haphazard, chaotic.

2) Supplement the information presented in the article on current technical, planning regulations in force in the region - by providing sources of information, literature, regional planning studies.

It appears from the article that references to local, regional architecture are required in new buildings, but are not followed and enforced. Are there any official design guidelines in local plans, spatial studies of the cities, of the region for this area? Maybe a comparison with the situation in other regions where regional architecture exists, in Romania and possibly a comparison with other countries of the former so called socialist block, which were undergoing transformation at a similar time? Are there any materials that present principles for the continuation of regional architectural features, construction in modern buildings, etc.?

It would be an important addition to this information and, of course, a commentary, due to the conclusions presented in the article and possible further research.

Author Response

Dear Reviewer,

Thank you for your valuable comments and suggestions we attentivelly followed in order to improve our manuscript.

Possible comparison studies with SEE countries would be a valuable scientific contribution to the literature that we have in view for future research. Unfortunatelly both in the case of Romania and for ex-socialist countries legislative initiatives to transpose architectural guidelines and studies into administrative practice are almost entirely missing at the moment or if they exist they are in an inception phase.

For more complete answers to all your comments please find the below attached word file entitled: author-coverletter-14497414.v1.docx

Kind regards,

The authors
